# Inflammatory and Prothrombotic Biomarkers, DNA Polymorphisms, MicroRNAs and Personalized Medicine for Patients with Peripheral Arterial Disease

**DOI:** 10.3390/ijms231912054

**Published:** 2022-10-10

**Authors:** Pavel Poredoš, Mišo Šabovič, Mojca Božič Mijovski, Jovana Nikolajević, Pier Luigi Antignani, Kosmas I. Paraskevas, Dimitri P. Mikhailidis, Aleš Blinc

**Affiliations:** 1Department of Vascular Diseases, University Medical Centre Ljubljana, 1525 Ljubljana, Slovenia; 2Department of Internal Medicine, Faculty of Medicine, University of Ljubljana, 1000 Ljubljana, Slovenia; 3Faculty of Pharmacy, University of Ljubljana, 1000 Ljubljana, Slovenia; 4Vascular Centre Nuova Villa Claudia, Via Flaminia Nuova 280, 00191 Rome, Italy; 5Department of Vascular Surgery, Central Clinic of Athens, 10680 Athens, Greece; 6Department of Clinical Biochemistry, Royal Free Hospital Campus (UCL), Department of Surgical Biotechnology, Division of Surgery and Interventional Science, University College London Medical School, University College London (UCL), Pond Street, London NW3 2QG, UK

**Keywords:** peripheral arterial disease, biomarkers, inflammation, prothrombotic, genetics, microRNA, personalized medicine

## Abstract

Classical risk factors play a major role in the initiation and development of atherosclerosis. However, the estimation of risk for cardiovascular events based only on risk factors is often insufficient. Efforts have been made to identify biomarkers that indicate ongoing atherosclerosis. Among important circulating biomarkers associated with peripheral arterial disease (PAD) are inflammatory markers which are determined by the expression of different genes and epigenetic processes. Among these proinflammatory molecules, interleukin-6, C-reactive protein, several adhesion molecules, CD40 ligand, osteoprotegerin and others are associated with the presence and progression of PAD. Additionally, several circulating prothrombotic markers have a predictive value in PAD. Genetic polymorphisms significantly, albeit moderately, affect risk factors for PAD via altered lipoprotein metabolism, diabetes, arterial hypertension, smoking, inflammation and thrombosis. However, most of the risk variants for PAD are located in noncoding regions of the genome and their influence on gene expression remains to be explored. MicroRNAs (miRNAs) are single-stranded, noncoding RNAs that modulate gene expression at the post-transcriptional level. Patterns of miRNA expression, to some extent, vary in different atherosclerotic cardiovascular diseases. miRNAs appear to be useful in the detection of PAD and the prediction of progression and revascularization outcomes. In conclusion, taking into account one’s predisposition to PAD, i.e., DNA polymorphisms and miRNAs, together with circulating inflammatory and coagulation markers, holds promise for more accurate prediction models and personalized therapeutic options.

## 1. Introduction

Peripheral artery disease (PAD) is a common manifestations of atherosclerosis affecting more than 200 million people worldwide [1,2]. The presence of PAD represents a powerful and independent risk of cardiac and cerebral ischemic events [3]. It is well-established that hypertension, smoking, diabetes mellitus and hypercholesterolemia play a major role in the initiation and development of atherosclerosis and cardiovascular (CV) events [4]. However, the prognostic potency of each of these factors in atherogenesis differs in various arterial beds and estimation of risk for atherosclerotic CV events based only on classical risk factors is often insufficient [4]. In ancient times when classical risk factors were less prevalent, atherosclerosis was already widespread as evidenced by Egyptian mummies [5].

Plaque formation is influenced by the expression of different genes [4,6,7]. Oxidized low density lipoprotein (LDL) particles which accumulate in the intima strongly modulate inflammation-related gene expression through various signaling pathways [8]. Epigenetic processes are also involved in cytokine expression that strongly affects atherogenesis [7]. MicroRNAs (mi-RNAs) which are single-stranded, non-coding RNAs that modulate gene-expression at the posttranscriptional level, have emerged as promising novel biomarkers for different aspects PAD, including its detection, prediction of progression and revascularization outcomes [9].

Biomarkers refer to a broad category of quantifiable biological characteristics that predict CV events. Besides providing diagnostic and prognostic information, some biomarkers might also be helpful in assessing the efficacy of therapeutic interventions [10].

The aim of this narrative review is to provide an overview of circulating biomarkers, which have already established their importance for the development of PAD, along with an overview of the emerging genetic biomarkers and miRNAs related to PAD. The three different, but to some extent interrelated, types of biomarkers could be used in the future for constructing mathematical prediction models for personalized medicine in PAD.

## 2. Inflammation and PAD

The role of inflammation in the atherogenic process was established as a basic pathophysiological process, which is responsible for the initiation, clinical manifestation and CV risk of atherosclerotic diseases [11]. The vascular inflammatory process is related to systemic inflammatory response and patients with preclinical or clinical atherosclerotic disease have increased circulatory inflammatory markers. Increased levels of inflammatory markers are associated with the development and progression of PAD, and the risk of developing cardiac and cerebrovascular ischemic events [12]. A number of cross-sectional and longitudinal studies demonstrated a close link between inflammation and PAD [13].

Inflammatory molecules are not simply markers of inflammation, they also play an active role in peripheral atherogenesis [14,15]. In a prospective case-control study in apparently healthy men enrolled in the Physicians Health Study (PHS) the relative risk of developing PAD increased significantly with each quartile of baseline C-reactive protein (CRP) and this increase was independent of other risk factors [16]. The PHS also showed that elevated levels of soluble intercellular adhesion molecule-1 (sICAM-1) are independently associated with the development of symptomatic PAD in men [17]. Similar findings were reported in the Women Health Study [18].

Among inflammatory biomarkers, the pro-inflammatory cytokine interleukin-6 (IL-6) was shown to be the strongest predictor of PAD and was independently associated with disease progression [13]. Circulating IL-6 levels significantly increase after exercise in patients with PAD, and higher IL-6 levels have been associated with lower functional capacity [19]. The Edinburg Artery Study also showed that interleukin-1 (IL-1) has an important predictive role in outcome of patients with PAD [20]. Interleukins, E-selectin and metalloproteinases predicted major events in patients with severe limb ischemia and allowed for the creation of a biomarker-model [21].

There are conflicting data on the role of interleukin-8 (IL-8). Some studies showed that PAD patients who were submitted to vascular surgical procedures had a higher production of IL-8 in polymorphonuclear leucocytes [22]. The anti-inflammatory interleukin-10 (IL-10), which is associated with reduced apoptosis of cells of the lipid core and thereby with the reduced risk of plaque rupture, has been related to reduced risk of developing atherosclerosis. However, the correlation between PAD and levels of IL-10 remains uncertain [23].

Several other inflammatory markers have been examined in a limited number of studies. In the Atherosclerotic Risk in Communities (ARIC) Study, monocyte chemoattractant protein-1 (MCP-1) was associated with the ankle-brachial index (ABI) [24]. Further, CD40 ligand was associated with an angiographic severity of PAD [25]. Serum osteoprotegerin was associated with a pathological ABI in a cohort study [26]. Wilson et al. reported that serum levels of of β_2_-microglobulin were higher in patient with PAD than in non-PAD patients and were independently associated with ABI [27].

### 2.1. Inflammatory Markers as Mediators of Harmful Effects of Other Risk Factors for PAD

Risk factors of atherosclerosis often trigger their atherogenic effects through an inflammatory mechanism (Figure 1). Cigarette smoking and diabetes mellitus, the strongest predictors of developing PAD, promote oxidative stress, which directly and indirectly stimulates inflammatory pathways [28].

Smoking is undoubtedly one of the most influential risk factors for PAD. The mechanisms associated with smoking include the activation of inflammation, dysregulation of lipid metabolism, increase of oxidative stress and endothelial dysfunction [29]. Smoking promotes inflammation through elevated white blood cell count, CRP, fibrinogen and von Willebrand factor which are elevated in patients with PAD [30]. Smoking also promotes activation of monocytes and production of various chemokines and cytokines [31]. 

Hypertension which is present in about 80% of PAD patients, also promotes inflammation [32]. Angiotensin II, which is involved in pathogenesis of arterial hypertension, elicits the production of reactive oxygen species and modifies the oxidation of LDL, stimulates the expression of vascular cell adhesion molecules and increases the expression of proinflammatory cytokines such as IL-6 [33] and dysregulates circulating miRNAs, which are associated with the presence of PAD and its progression [29]. Smoking cessation can correct abnormalities related to smoking including vascular inflammation, dyslipidemia, endothelial dysfunction, arterial stiffness and insulin resistance, but the success rates of smoking cessation are relatively low [34].

Diabetes also ranks among the strongest risk factors for PAD. Several biomarkers of PAD were identified in diabetic patients. A significant association was shown between HbA1c levels and the incidence of PAD [35]. Ischemia-modified albumin in type 2 diabetic patients which is associated with HbA1c, was also shown to be a risk marker of PAD [36]. Further, copeptin, B-type of natriuretic peptide and cystatin C is associated with the incidence of symptomatic PAD [37]. In diabetic patients, amputations re-amputations represent frequent complication. Guelcu et al. showed that CRP, together with lower albumin, higher HbA1c, and higher creatinine levels is associated with poor prognosis and re-amputation [38]. Diabetes induces dysregulation of miRNA expression that is associated with the development of macrovascular complications, including PAD [39] and dysregulates miRNAs expression related to atherosclerosis [40].

Dyslipidemia also plays a pivotal role in the activation of inflammatory pathways, increasing the production of inflammatory cytokines, mainly tumor necrosis factor alpha (TNF-α) and IL-6 [41] and inducing miRNA dysregulation [40].

### 2.2. Effects of Preventive and Therapeutic Measures on Inflammatory Biomarkers

Determination of inflammatory biomarkers can be used as an indicator of effects of preventive and therapeutic measures in patients with atherosclerotic disease including PAD.

#### 2.2.1. Physical Exercise 

Patients with PAD regularly experience ischemia of the tissue distant to arterial occlusions during exercise. The transient exercise-induced leg ischemia is related to increased release of inflammatory markers [42] and impairs vasodilator function of distant arteries in correlation with increased circulatory levels of interleukins, particularly IL-6 [43]. This could be one of the reasons that because of repeated ischemia-related release of inflammatory markers, PAD patients experience advanced systemic atherosclerosis including coronary heart disease (CHD) and other CV disease (CVD).

In contrast to the acute deleterious effects of ischemia, regular moderate exercise training not only improves walking capacity but also decreases vascular and inflammatory biomarkers [44]. Chronic exercise improves anti-oxidant capacity and decreases inflammatory response without increasing oxidative stress in symptomatic PAD patients [42,44]. Acute exercise usually leads to robust inflammatory response mainly characterized by the mobilization of leukocytes and an increase in circulating inflammatory mediators produced by immune cells and directly from active muscle tissue [44]. 

Moderate physical exercise training results in improvements in systemic inflammation, evident by reduction in acute phase proteins [45]. Therefore, repeated moderate physical exercise in patients with intermittent claudication reinforces antioxidant capacity, reduces oxidative stress and inflammation and regulates the immune response [46]. 

Overall, physical training is one of the most effective treatment options for PAD patients, which prevents progression of local disease and CV events. These benefits are most probably based on reducing the inflammatory response and improving immune function.

#### 2.2.2. Statins

The anti-inflammatory properties of statins represent one of the basic anti-atherosclerotic mechanisms, involving a reduction in the release of CRP, chemokines, cytokines, and adhesion molecules [47]. Furthermore, statins inhibit the transendothelial migration of leukocytes [48]. In addition, statins have been shown to decrease the number of inflammatory cells in atherosclerotic plaques and to possess other anti-inflammatory properties [47]. This could be the consequence of inhibition of adhesion molecules or cytokines (IL-6, IL-8), which are involved in the accumulation of inflammatory cells. The importance of inhibition of inflammation by statins was shown in the Justification for the Use of Statins in Prevention: an Intervention Trial Evaluating Rosuvastatin (JUPITER) study [49]. Besides stabilization and regression of atherosclerotic plaques, statins were shown to reduce inflammation [lowering of levels of CRP, fibrinogen, neutrophils], which, in patients with PAD correlates with improved survival [50,51]. Therefore, an important mechanism by which statins improve outcomes in atherosclerotic patients, including PAD, may be the reduction of vascular inflammation [52].

Among patients with PAD in the National Veterans Affairs cohort, any statin use reduced mortality and high intensity statin use also reduced limb amputations [53,54].

##### 2.2.3. “Novel” Anti-inflammatory Agents and Approaches 

Several novel anti-inflammatory agents are being tested in prevention of atherosclerosis. Canakinumab, a monoclonal anti-IL-1beta antibody effectively reduced atherosclerotic CV events in the Canakinumab Antiinflammatory Thrombosis Outcome Study (CANTOS) study independent of lipid-lowering, also reduced cancer mortality, especially of lung cancer, but increased the incidence of fatal infections [55]. On the other hand, low-dose methotrexate did not result in a clinical benefit in very high risk patients with previous myocardial infarction or multivessel coronary disease who also had either type 2 diabetes or metabolic syndrome [56]. 

Colchicine is an ancient herbal drug with powerful anti-inflammatory potency. Colchicine reduces the levels of pro-inflammatory cytokines and stabilizes the coronary plaques, leading to a reduction of recurrent coronary events after acute coronary syndromes and better outcomes in patients with chronic coronary disease [57,58]. The efficacy of colchicine in reducing cardiovascular and limb events in patients with symptomatic PAD is being tested in the on-going LEADER-PAD (Low dose Colchicine in Patients with peripheral Artery DiseasE) trial (ClinicalTrials.gov: NCT04774159).

The use of immunosuppressive drugs targeting chronic inflammation could assume an important role in the future. Further, drugs with anti-inflammatory capacity, currently used for other indications, might be reprogrammed for use in PAD. Among them, sodium-glucose cotransporter 2 inhibitors and glucagon-like peptide-1 receptor agonists are particularly interesting [59]. Overall, new anti-inflammatory drugs and approaches are already on the horizon. It seems likely that in the future anti-inflammatory treatment will be guided by a personalized approach, based on the individual risk profile. 

## 3. Procoagulant and Fibrinolytic Markers of PAD

PAD is often accompanied by a prothrombotic state [60]. Plasminogen activator inhibitor-1 (PAI-1) levels were higher both at rest and after exercise in PAD patients compared with healthy subjects [61]. Circulating prothrombin fragments 1 + 2 and thrombin-antithrombin complex (TAT) which are markers of thrombin generation were elevated in PAD patients [62]. In addition, patients with critical limb ischemia (CLI) had even higher levels of TAT [63]. Tissue factor (TF) and von Willebrand factor which are at the top of the coagulation cascade were higher in PAD patients than controls, and TF antigen was higher in CLI than in other stages of PAD [62]. Patients with PAD also exhibited elevated concentration of platelet factor 4 [62]. Fibrinogen, an acute phase reactant regulated by IL-6 has been reported to be higher in PAD patients than in healthy controls and fibrinogen concentration increased with severity of disease [63]. Fibrinogen concentration also predicted mortality in PAD [64].

In addition to hypercoagulability, platelet hyperactivity is present in patients with PAD and is not completely attenuated by antiplatelet therapy [65]. However, antiplatelet therapy reduces vascular events and death in patients with PAD [66]. Patients with PAD are characterized by increased levels of P-selectin and CD40 ligand [67,68].

Low mean platelet volume, especially ≤10.2 fL was associated with progression to critical limb ischemia in patents with PAD [69].

Antiplatelet therapy with aspirin or clopidogrel decreases cardiovascular events in patients with PAD [70]. It has been shown in the COMPASS trial that the combination of low-dose rivaroxaban [2.5 mg bid] with aspirin is superior over aspirin alone for reducing cardiovascular and limb events in symptomatic PAD patients [71]. However, since the COMPASS trial was stopped early due to overall efficacy, the confidence intervals for major adverse limb events including major amputation were less precise than desired [71]. Which subpopulation of PAD patients may benefit the most is not known. The personalized approach implementing individual risk score might be beneficial for patient stratification for current and possible future antiplatelet/anticoagulation treatment of PAD.

## 4. Differences in Circulating Biomarkers of PAD and CHD

Since atherosclerosis is a systemic disease it is unlikely that plasma biomarkers would only be specific for PAD as opposed to CHD. However, there are some differences in biomarkers between CHD and PAD which are caused by different endothelial and vascular smooth muscle gene expression in different vascular territories [72]. Differences in vascular reactivity between the peripheral and coronary arteries are a consequence of differences in the expression of cell surface receptors and signaling pathways leading to differences in expression of endothelial chemokines and adhesion molecules exists [73]. It has been hypothesized that because of phenotypic differences between peripheral and coronary vascular cells, the ischemic response and release of biomarkers differs between the vascular territories [74]. 

A substantial subset of circulating biomarkers—C-terminal endothelin-1 (CT-proET-1), N-terminal prosomatostatin ([NT-proSST), midregional proatrial natriuretic peptide (MR-proANP), procalcitonin (PCT), and copeptin—were prospectively evaluated for their role in PAD development and mortality in 3618 men and 1542 women in the Malmö Preventive Project [75]. Increased levels of CT-proET-1, NT-proSST, and MR-proANP were independently associated with incident PAD, whereas all the vasoactive biomarkers were independently associated with mortality during the 11-year follow-up [75]. 

## 5. Genetic Markers of PAD

Gene variants and gene-environment interactions have a moderate, but significant role in the development and progression of PAD [76]. Historically, twin studies and family-based estimates were used to quantify heritability, i.e., the amount of phenotypic variation in a population that is attributable to genetic differences between individuals [76]. The estimated heritability of PAD; i.e., the amount of phenotypic variation in a population that is attributable to genetic differences between individuals within the population, is 0.2–0.3, which is similar to other phenotypes of atherosclerotic disease [76].

### 5.1. Early Attempts at Identifying DNA Loci Associated with PAD 

Linkage analysis, a statistical estimate of whether genetic markers co-segregate among related individuals with the disease, initially found a region on chromosome 1p31 that was associated with PAD in residents of Iceland [77], but another study in African Americans and Non-Hispanic whites did not replicate this result [78]. 

Candidate gene studies, using case-control methodology to investigate differences in allele frequency between individuals with and without PAD, identified several biological pathways involved in atherosclerosis and PAD, including leukocyte adhesion [79], coagulation [80], and inflammation [81]. A small study of symptomatic PAD patients implicated activation of the Notch signaling pathway, a major regulator of vascular smooth muscle cells and macrophages, through the Delta like 4 ligand to the gene expression of »inflamed plaque« leading to unfavorable progression of PAD [82]. The largest candidate gene study on PAD was performed by the Candidate Gene Association Resource consortium evaluating approximately 50,000 gene variants in >29,000 individuals [83]. Two variants, rs2171209 in *SYTL3* and rs290481 in *TCF7L2* were significantly associated with ABI in European Americans, but not in African Americans. The association between the two single-nucleotide polymorphisms (SNPs) that met experiment-wide significance for ABI in European Americans could not be replicated in an additional study involving 13,524 Europeans and Americans of European ancestry [83]. In general, candidate gene studies were plagued by the arbitrary selection of candidate genes and frequent failure of replicating results in other cohorts [76]. For example, the presence of *NURR1* variants rs13428968 and rs12803 was associated with restenosis/re-occlusion rate after femoropopliteal percutaneous angioplasty [84], but in another population, the presence of those variants was not associated with either PAD or adverse CV events [85].

### 5.2. Genome-Wide Association Studies (GWAS) in Identifying DNA loci Associated with PAD

During the last decade, with the availability of high throughput and relatively low-cost genotyping, GWAS became the dominant methodology for studying genetics of PAD. GWAS investigate the association between the disease and genetic variants across the entire genome. [76]. GWAS methodology accounts for multiple testing by dividing the widely accepted statistical significance threshold (*p* < 0.05) by the number of independent segments in the human genome (10^6^), with the quotient (*p* < 5 × 10^−8^) representing the accepted GWAS significance threshold of [76]. 

GWAS investigated PAD-related traits such as smoking [86], coronary artery disease [87], and abdominal aortic aneurysm [88], and demonstrated suggestive PAD associations for the loci *9p21/CDKN2B* and *CHRNA3*.

Table 1 shows DNA polymorphisms that have been associated with PAD in large GWAS.

In a Japanese population, a suggestive association with PAD was observed for the polymorphism rs1902341 in the intron of *OSBPL10* (odds ratio (OR), 1.31, *p* = 4 × 10^−7^) [89], and significant associations were found for rs9584669 in *IPO5/RAP2A* (OR = 0.58, *p* = 6.8 × 10^−14^) and rs6842241 in *EDNRA* (OR, 0.85, *p* = 5.3 × 10^−9^) [90]. In a meta-analysis, rs10757269 on chromosome 9 near *9p21/CDKN2B* demonstrated the strongest association with ABI, *p* = 2.46 × 10^−8^ [91].

Very large case-control studies were carried out with the use of electronic health records to determine the presence or absence of PAD. Approximately 32 million DNA sequence variants were tested for association with PAD in 31,307 cases and 211,753 controls in the Million Veteran Program population in cooperation with the United Kingdom Biobank, verifying the results in an independent sample of 5117 PAD cases and 389,291 controls [92]. In total, 19 PAD-associated variants were identified, including the previously identified *9p21* locus and 18 new variants [76,92]. Association with CHD was found in 14 PAD risk variants, and association with large artery stroke (LAS) in 12 variants [76], indicating that some of the PAD risk may have been driven by comorbidity or shared causal pathways for atherosclerosis. Possible biological mechanisms were identified only for some of the identified DNA polymorphisms (Table 1). Variants that were uniquely associated with PAD included in the *CHRNA3* locus, known to be associated with nicotine dependence and the prothrombotic Factor V Leiden (rs6025) [76]. 

**Table 1 ijms-23-12054-t001:** DNA polymorphisms associated with peripheral arterial disease in large genome wide association studies. Genes for variants outside the transcript boundary of a protein-coding gene are shown with nearest candidate gene in parentheses. The magnitude of effect is small if the odds ratio for PAD in the presence of the DNA polymorphism is estimated at 1.01–1.10, and moderate if the odds ratio is 1.11–1.30. (rsID—reference single nucleotide polymorphism cluster identifier).

Chromosome	Gene/Locus	rsID	Magnitude of Effect	Proposed Mechanism	Brief Annotation	Ref.
1	*CELSR2/SORT1*	rs7528419	small	lipoprotein metabolism	3′ untranslated region variant	[92]
1	*F5*	rs6025	moderate	thrombosis	missense variant (Factor V Leiden)	[92]
3	*OSBPL10*	rs1902341	moderate	lipoprotein metabolism	intron variant	[89]
4	*EDNRA*	rs6842241	moderate	vasoconstriction/inflammation	gene variant	[90]
6	*LPA*	rs118039278	moderate	lipoprotein metabolism	intron variant	[92]
6	*(HLA-B)*	rs3130968	small	?	regulatory region variant	[92]
7	*(HDAC9)*	rs2074633	moderate	cell cycle regulation	regulatory region variant	[90]
7	*(HDAC9)*	rs2107595	small	?	regulatory region variant	[92]
7	*[IL6]*	rs4722172	small	inflammation	intergenic variant	[92]
8	*LPL*	rs322	small	lipoprotein metabolism	intron variant	[92]
9	*(CDKN2B)*	rs10757269	moderate	cell cycle regulation	regulatory region variant	[91]
9	*ABO*	rs505922	small	?	intron variant	[92]
9	*CDKN2B-AS1/9p21*	rs1537372	small	?	intron variant	[92]
10	*TCF7L2*	rs7903146	small	diabetes	intron variant	[92]
11	*MMP3*	rs566125	small	?	intron variant	[92]
11	*CREB3L1*	rs7476	small	?	3′ untranslated region variant	[92]
12	*PTPN11*	rs11066301	small	?	intron variant	[92]
12	*RP11-359M6.3*	rs4842266	small	?	upstream gene variant	[92]
13	*IPO5*	rs9584669	moderate protective	lipoprotein metabolism	regulatory region variant	[90]
13	*COL4A1*	rs1975514	small	?	intron variant	[92]
14	*SMOC1*	rs55784307	small	?	downstream gene variant	[92]
15	*CHRNA3*	rs10851907	small	smoking/nicotine dependence	upstream gene variant	[92]
17	*LOC732538*	rs62084752	small	?	upstream gene variant	[92]
19	*(LDLR)*	rs138294113	small	lipoprotein metabolism	intergenic variant	[92]

Mendelian randomization studies using GWAS methodology have confirmed the association of PAD with hemostatic factors, i.e., genetically increased factor VIII activationand von Willebrand factor concentration [93], and with altered lipoprotein metabolism i.e., genetically determined extra-small very low density lipoproteins [94]. Genetically determined arterial hypertension has been associated with CHD and to a lesser extent with PAD [95]. Genetic liability to smoking has been associated with CHD, large artery stroke and most notably with PAD [76].

Most of the risk variants associated with PAD involved noncoding regions which are believed to alter transcription of nearby genes [96]. Understanding the interplay between noncoding and coding regions of the genome is a big challenge for the future, as is constructing accurate polygenic risk scores and better understanding of gene-environment interactions [76]. 

Structural variants, such as inversions, duplications, translocations, large deletions, and large insertions have been implicated in altered atherosclerotic disease risk [76], with solid evidence for CHD [97], but still lacking evidence for PAD.

In summary, genetic predisposition significantly, albeit moderately affects risk factors for PAD and PAD itself. Genetic basis for altered lipoprotein metabolism, diabetes, arterial hypertension, smoking, inflammation and thrombosis has been implicated in pathophysiological pathways leading to PAD. However, most of the risk variants for PAD are located in noncoding regions of the genome and their influence on gene expression remains to be explored.

### 5.3. Somatic Mutations in Hematopoietic Stem Cells

Age-related somatic mutations in hematopoietic stem cells may result in clonal hemopoiesis of indeterminate significance (CHIP), which increases the risk of hematologic malignancies, CVD and overall mortality [98,99]. At present, the importance of CHIP for PAD remains unknown.

## 6. Micro RNAs and PAD

MicroRNAs (miRNAs) are small (up to 25 nucleotides long), endogenous, single-stranded, noncoding RNAs that modulate gene expression at the post transcriptional level [9,100] miRNAs act mostly by directly binding to the 3′ untranslated region (3′ UTR) of target messenger RNA (mRNA) sequences, thus inhibiting translation and/or promoting the degradation of target mRNA. In rare cases miRNAs could also promote mRNA translation [9,100]. Extensive research of miRNAs biology revealed that some of their characteristics could be used as biomarkers. miRNAs circulating in human serum and body fluids are remarkably stable at different pH and temperature levels [101,102]. Binding to high- and low-density lipoproteins (HDL/LDL) and RNA-binding proteins or their incorporation in membranous vesicles (exosomes) and apoptotic bodies makes miRNAs resistant to endogenous RNAse activity [103]. In contrast to protein biomarkers, miRNAs have less complex structure, are not susceptible to posttranslational modifications and can be detected with high sensitivity and specificity using sequence-specific amplification [102]. Furthermore, miRNAs expression is regulated in a tissue- and disease-specific manner, resulting in different patterns of miRNAs expression in different CVD [CHD, myocardial infarction, heart failure, PAD]. In the vasculature, different miRNAs regulate the post transcriptional expression of genes involved in structural remodeling, inflammation, angiogenesis, atherosclerosis, in-stent restenosis, and thrombosis [104]. All these characteristics suggested that miRNAs may be useful novel biomarkers for a wide variety of diseases. 

### 6.1. miRNAs as Diagnostic Markers for PAD

Several studies on patients with PAD have revealed clusters of miRNAs with deregulated expression that could serve as a diagnostic biomarker for PAD. Stather et al. performed miRNA transcriptomic analysis in patients with PAD and identified 12 miRNAs that could be of diagnostic value: let 7e, miR-15b, -16, -20b, -25, -26b, -27b, -28-5p, -126, -195, -335, and -363 [105]. Most of them were downregulated in plasma and modulate the expression of genes with well-established role in the pathophysiology of CVD: intercellular adhesion molecule 1 (ICAM1), vascular cell adhesion protein (VCAM), transforming growth factor beta receptor 2 (TGFBR2), selectin, integrin, nuclear factor kappa B (NFkB). Signorelli et al. found up-regulated miRNA-130a, miRNA-27b, and miRNA-210 in serum samples of PAD patients [106]. These miRNAs play an important role in the modulation of oxidative stress [106]. Pereira-da-Silva et al. performed a systematic review and found decreased levels of let 7e, miRNA-27b, miRNA-130a, and miRNA-210 to be specific for PAD [107]. Transcriptomic analysis in peripheral blood mononuclear cells (PBMCs), an important element of inflammation in atherosclerotic process, revealed deregulated expression of 26 microRNAs and 14 genes in patients with PAD [108]. 

miRNAs, identified as being deregulated in patients with PAD, modulate different aspects of atherosclerotic disease: endothelial cell function, inflammation, oxidative stress, angiogenesis, extracellular matrix composition, and cholesterol metabolism [107]. In addition, the authors found up-regulated miRNA-21 and down-regulated miRNA-30, miRNA-126, and miRNA-221-3p to be most commonly deregulated in atherosclerotic disease, although this finding was not specific for PAD. It is not surprising that CHD, carotid disease and PAD share the same cluster of deregulated miRNAs as atherosclerotic disease is common denominator for all the arterial beds. On the other hand, some miRNAs are specific for particular arterial beds as some factors and triggers, like shear stress, differentially contribute to the development and progression of atherosclerosis [107].

### 6.2. miRNAs and PAD Severity

Hypoxia, as a result of PAD, is associated with mitochondrial dysfunction in skeletal muscle cells that impairs energy metabolism and is partially responsible for functional limitations in patients with PAD. Ischemia was reported to change the expression of miRNAs relevant to mitochondrial function: miR-210, miRNA-499 [109,110]. Circulating miRNA-210 and miRNA-124, both related to oxidative stress and angiogenesis, were found to have an inverse correlation with claudication distance and the ABI [111,112]. Levels of miRNA-548j-5p that promotes angiogenesis were decreased in patients with PAD compared to healthy control subjects [113].

Moreover, exercise therapy in patients with PAD is associated with symptoms relief and is, at least partially, associated with the restoration of mitochondrial function [114]. Exercise training was reported to be associated with reduced expression of miRNA-494 and miRNA-696 [115,116]. Increased expression of miRNA-1827 was associated with CLI, while miRNA-4739 was associated with CLI in patients with diabetes [117,118].

#### miRNAs in Experimental Models of PAD

In an experimental model of limb ischemia in diabetic mice, inhibiting miRNA-29a upregulated the expression of the ADAM12 gene, coding for the disintegrin and metalloproteinase domain-containing protein 12, which improved perfusion recovery [119]. In mice with glucose intolerance, after inducing hind limb ischemia, administration of unacylated ghrelin upregulated miRNA-126, decreasing the level of VCAM-1 and the number of infiltrating inflammatory cells in ischemic muscle, potentially rescuing ischemic muscle from necrosis [120]. On the other hand, ghrelin-knockout mice exhibited an impaired response to hind limb ischemia due to decreased levels of miRNA-126 and miRNA-132 [121]. The interleukin-21 receptor (IL-21R) was associated with good recovery from hind-limb ischemia in a mouse model, and IL-21R was upregulated by miRNA-30b [122]. Intra-muscular delivery of miRNA-93 in genetically modified mice subjected to hind-limb ischemia increased angiogenesis, arteriogenesis, and the extent of perfusion by modulating macrophage polarization [123].

### 6.3. miRNAs as Markers of In-Stent Restenosis

In-stent restenosis after peripheral artery angioplasty resulted from the generation of neointima, as a result of the proliferation and abluminal migration of vascular smooth muscle cells (VSMC) and extracellular matrix deposition [124]. Yuan et al. reported higher expression of miRNA-320a and miRNA-572 in patients with in-stent restenosis following peripheral artery angioplasty [125]. Of note, miRNAs previously reported to be associated with in-stent restenosis following coronary angioplasty were undetectable in almost half of the patients tested, suggested specific miRNA expression patterns in CHD and PAD [125]. Other studies reported miR-140-3p’s and miR195 expression to be downregulated in patients with in-stent restenosis after peripheral artery angioplasty [126,127]. Downregulation of miRNA-195 provided additional prognostic value for in-stent thrombosis beyond traditional risk factors including age, sex, smoking, diabetes, hypertension, and hyperlipidemia [127]. These findings suggest potential role of miRNAs in risk stratification and prediction of adverse ischemic events after stent implantation in patients with PAD, which could be a step towards more personalised approach in PAD treatment. 

### 6.4. miRNAs as Predictors of CV Events

Alterations in miRNAs expression could also predict CV events. Stojkovic et al. found that up-regulated miRNA-142 in PAD patients after femoral bypass surgery independently predicts future coronary events and vascular surgical procedures, toe or leg amputations, within one year of follow-up [127]. Vegter et al. found that several miRNAs, related to PAD, could predict future CV complications in patients with heart failure [128]. Badacz et al. identified miRNA-1-3p, miRNA-16-5p and miRNA-122-5p as possible risk factors for secondary ischemic events in patients with carotid and/or coronary artery disease [129]. In patients with PAD following femoral artery bypass surgery, plasma levels of miRNA-142, miRNA-223, miRNA-155, and miRNA-92a predicted CV events [130]. The miRNA-34 family members miRNA-34a, miRNA-34b and miRNA-34c were additively involved in human arterial disease, affecting the presence of coronary plaques, the number of affected vascular beds and aortic stiffening [131].

In summary, miRNAs appear to be promising novel biomarkers for different aspects of PAD including detection, risk stratification, prediction of progression, complications and revascularization outcome. Furthermore, miRNAs could serve as potential therapeutic targets, representing valuable tools for the development and implantation of personalised approach for patients with PAD. Changes in miRNAs expression, specific for a particular CVD, could suggest novel signalling pathways and mechanisms that play an important role in the complex pathophysiology of CVD thus improving our understanding of the process and risk factors that significantly contribute to it. Notably, the idea of specific pattern of miRNA expression for atherosclerotic disease in different arterial territories seems attractive, but the evidence needed for establishing the most accurate and reliable panel for particular arterial territory is still lacking.

## 7. Discussion

### 7.1. Integrating an Individual’s Circulating Biomarkers, DNA Polymorphisms and miRNA—A Possible Step Forward in Personalized Medicine for PAD

CV medicine is currently lagging behind oncology in personalized selection of treatments, but there is an emerging need for appropriate selection of patients for treatments that are already available or will be accessible in the near future [132,133]. This is especially true of new anti-inflammatory treatments, which will very likely become an important part of the management of selected PAD patients and atherosclerosis treatment in general [55,132,134,135].

Several characteristics of individuals, such as their genetics, epigenetics, demographic characteristics, lifestyle, environment exposures, body composition, traditional risk factors, and other comorbidities influence the course of PAD [132,136]. All these parameters could in the future be used to create a personalized approach. Since inflammation biomarkers reflect the individual’s ongoing pathophysiological process in all stages of PAD they seem strong candidates for experimental evaluation in complex scoring algorithms predicting the incidence and clinical course of PAD. Comprehensive electronic health records of large numbers of people together with deep-learning methodology would be required to achieve this goal. This type of collaboration between computer scientists and physicians has already shown considerable promise, as described in the review by Saenz-Pipaon et al. [137]. Also, research is ongoing to define and evaluate mathematical scoring algorithms of polygenic risk scores [PRSs] to integrate genotypes into the prediction of likelihood of developing atherosclerosis [138,139]. Less is currently known about the scoring algorithms for miRNAs and inflammatory/coagulation biomarkers. Integrating an individual’s susceptibility for PAD based on DNA polymorphisms and miRNA with the individual’s circulating markers of inflammation and activated coagulation (representing ongoing activity of the underlying mechanisms) in mathematical prediction models might be one of the possible personalized approaches (Figure 2). Undoubtedly, a significant gap is present in the use of personalized medicine in PAD [136], but with rapid progress of information technology and deep learning this gap might start shrinking. 

### 7.2. Conclusions

Several circulating biomarkers, including markers of inflammation and hemostasis are associated with the incidence and clinical course of PAD. Among inflammatory markers, IL-6 is the strongest predictor of PAD and is independently associated with disease progression. CRP, several adhesion molecules, CD40 ligand, and osteoprotegerin are also associated with the presence and progression of PAD. Circulating thrombin fragments and thrombin- antithrombin complex are elevated in patients with PAD, especially in CLI.

Genetic basis for altered lipoprotein metabolism, diabetes, arterial hypertension, smoking, inflammation and thrombosis has been implicated in pathophysiological pathways leading to PAD. However, most of the DNA risk variants for PAD are located in noncoding regions of the genome and their influence on gene expression remains to be explored.

MiRNAs are promising novel biomarkers for PAD detection, risk stratification, prediction of disease progression, complications and revascularization outcomes.

With rapid advances in information technology and deep learning methodology, the established circulating biomarkers of PAD together with DNA polymorphisms and miRNAs offer hope for more accurate prediction models and progression toward personalized treatment of PAD.

## Figures and Tables

**Figure 1 ijms-23-12054-f001:**
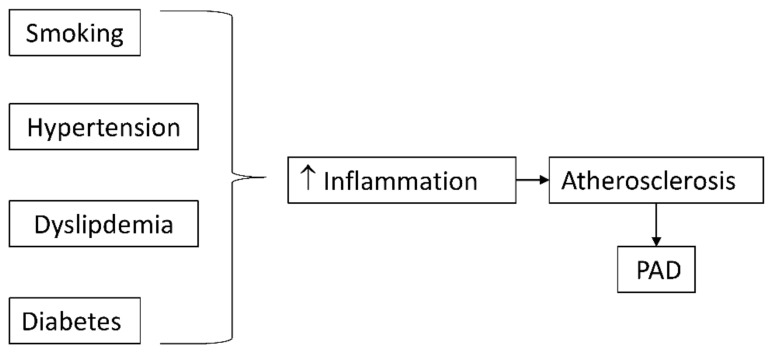
Classical risk factors for atherosclerosis promote inflammation which accelerates atherosclerosis and peripheral arterial disease (PAD).

**Figure 2 ijms-23-12054-f002:**
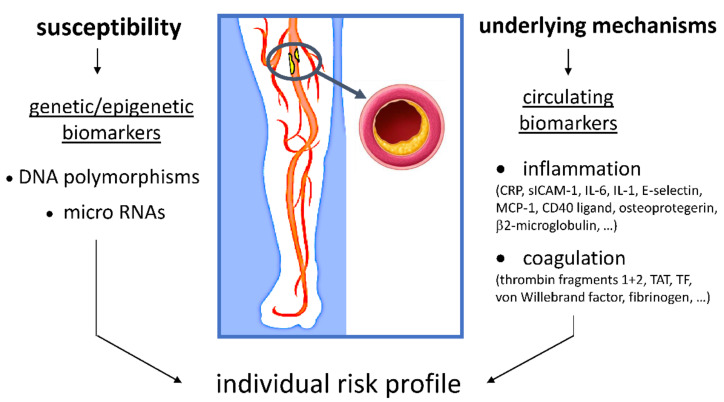
Schematic approach to integrating susceptibility for PAD, based on DNA polymorphisms and micro RNA, with ongoing inflammation and activated coagulation into an individual’s personalized risk. (CRP –C-reactive protein, sICAM-1 –soluble intercellular adhesion molecule-1, IL-6—interleukin-6, IL-1—interleukin-1, MCP—monocyte chemoattractant protein-1, TAT- thrombin antithrombin complex, TF—tissue factor).

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
