# Peer review of "Inflammatory and Prothrombotic Biomarkers, DNA Polymorphisms, MicroRNAs and Personalized Medicine for Patients with Peripheral Arterial Disease"

_ijms, 2022, doi:10.3390/ijms231912054_

Round 1

Reviewer 1 Report

Info in 2.1 is well-know; should be presented in a table to attract readers.

2.2.3 should be extended; f.e. the study of canakinumab cited shows different results that can be commented deeper.

The MS only reflects a general idea of the topics and is too general. Authors must argue about results and limitations of some studies cited; f.e. reference 70

L-383 to L-388 please cite support bibliography

Author Response

Dear Editor,

We thank the reviewers for their thoughtful comments on our manuscript IJMS-1936807 “DNA polymorphisms, microRNAs, and personalized medicine for patients with peripheral arterial disease”.

According to the suggestions of the reviewers, we have made the following changes:

Reviewer 1

  • Info in 2.1 is well-known; should be presented in a table to attract readers.

A brief scheme (Fig. 1) has been added to section 2.1 to illustrate the point that classical risk factors promote inflammation which in turn accelerates atherosclerosis and PAD as one of its major clinical manifestations. We opted for a simple scheme rather than for a comprehensive table in order to follow the advice of attracting readers.

  • 2.2.3 should be extended; f.e. the study of canakinumab cited shows different results that can be commented deeper.

More information has been added on the CANTOS study in lines 202-204:  Canakinumab, a monoclonal anti-IL-1beta antibody effectively reduced atherosclerotic CV events in the Canakinumab Canakinumab Antiinflammatory Thrombosis Outcome Study (CANTOS) study independent of lipid-lowering, also reduced cancer mortality, especially of lung cancer, but increased the incidence of fatal infections.  

  • The MS only reflects a general idea of the topics and is too general. Authors must argue about results and limitations of some studies cited; f.e. reference 70

The main limitation of the COMPASS trial regarding PAD patients has been added (lines 245-247): However, since the COMPASS trial was stopped early due to overall efficacy, the confidence intervals for major adverse limb events including major amputation were less precise than desired [70].

  • L-383 to L-388 please cite support bibliography

Our apologies. The missing reference [103] for statements, now in lines 391-396, has been added.

Reviewer 2

  • In Figure 1. Is it possible to list the main circulating biomarkers of inflammation and coagulation? It would be very useful.

We have listed the most important inflammatory markers and coagulation markers in the figure, now Fig.2.

  • Which markers of atherosclerosis are similar with the markers of PAD?

Similarities and differences in circulating biomarkers of PAD and coronary heart disease, the most extensively studied atherosclerotic disease, are discussed in section 4 (lines 252-269).

Hoping that the revised manuscript is now suitable for publication in the International Journal of Molecular Sciences.

On behalf of all co-authors,

With best regards,

Ales Blinc

Reviewer 2 Report

In the review Poredos and co-authors summarises our knowledge about the predisposition factors to periphery vascular disease (PAD).  This is a comprehensive review that provides an up-to-date appraisal of the literature. The authors focused on the inflammatory and prothrombotic biomarkers, DNA polymorphisms, and microRNAs, which can help in understanding the pathomechanisms of PAD. This knowledge may lead toward personalized treatment of PAD. The authors have carried out a thorough work based on a large number of own and international publications. The review is well designed and good written.

Comments:

-           In Figure 1. Is it possible to list the main circulating biomarkers of inflammation and coagulation? It would be very useful.

-          Which markers of atherosclerosis are similar with the markers of PAD?

Author Response

(The authors gave the same response as above.)
